# Assessing the role of peer education in improving clinical and patient-reported outcomes in adults with chronic kidney disease: A scoping review protocol

Anna Winterbottom[1]*, Jyoti Baharani[2], Hilary L. Bekker[3]

1 Visiting Associate Professor, Leeds Institute of Health Sciences, School of Medicine, Senior Health Services Researcher, Adult Renal Services, St James University Hospital, University of Leeds, Leeds, United Kingdom, 2 Consultant Physician and Nephrologist, Birmingham Heartlands Hospital, University Hospital Birmingham, United Kingdom, 3 Professor of Medical Decision making, Leeds Institute of Health Sciences, School of Medicine, University of Leeds, Leeds, United Kingdom

* a.e.winterbottom@leeds.ac.uk

## Abstract

### Purpose

This paper describes a protocol for a scoping review examining the role of peer education programs for people with chronic kidney disease across diverse healthcare settings and cultural contexts.

### Method

The protocol is guided by the Joanna Briggs Institute evidence synthesis guidance, and reported here following the Preferred Reporting Items for Systematic Reviews and Meta-Analyses for scoping reviews (PRISMA-P) guideline. Based on the population, intervention, comparator, and outcome (PICO) framework, our research questions are: 1) What models of peer education are used to supplement pre-treatment education in adults with chronic kidney disease? 2) What specific clinical and patient-reported outcomes are impacted by peer education in adults with chronic kidney disease? 3) What are the core components of peer education interventions for people with chronic kidney disease? 4) Are there variations in the need for, and experience of, peer education by patient characteristics (including ethnicity, socio-economic status, age, gender, disease stage)? The review process includes: 1) development of a comprehensive search strategy and inclusion criteria, developed iteratively with an information specialist to balance sensitivity (capturing all relevant studies) and specificity (minimizing irrelevant retrievals); 2) data extraction; 3) charting the data and quality assessment using the Mixed Methods Appraisal Tool (MMAT) for diverse study designs and the Joanna Briggs Institute (JBI) critical appraisal checklists for specific and 4) data synthesis and analysis. A data extraction form will

**Data availability statement:** No datasets were generated or analysed during the current study. All relevant data from this study will be made available upon study completion.

**Funding:** This project is funded by Kidney Research UK. The funders did not have any role in study design; and will play no role in data collection, analysis, interpretation of data, writing the review, and the decision to submit the review for publication.

**Competing interests:** The authors have declared that no competing interests exist.

elicit identical information from each study meeting the inclusion criteria. Data analysis will be determined by findings, and is likely to include quantitative summaries organized by outcome categories (clinical outcomes, patient-reported outcomes, decision-making quality) and narrative synthesis. Where heterogeneity precludes meta-analysis, findings will be synthesized descriptively using tables and narrative summary.

## Conclusion

This scoping review will synthesize current evidence from diverse international healthcare contexts, and identify research gaps, to provide guidance for services on defining, developing and implementing peer education interventions that meet the needs of people with chronic kidney disease globally, with attention to contextual factors that may influence transferability across healthcare systems.

## Background

Chronic kidney disease (CKD) is a severe, long-term condition in which the kidneys gradually lose their ability to function [1,2]. As the disease progresses to kidney failure, patients must consider various treatment options, including dialysis, kidney transplantation, or conservative kidney management. Engaging patients in shared decision-making regarding these treatment pathways is crucial [3,4]. However, there are no standardized guidelines to assist healthcare professionals in preparing patients to make these critical decisions [5]. As a result, there is substantial variation in how decision-making support is structured across healthcare services, the types of educational resources provided, and how patients experience the decision-making process. Research indicates that patients with CKD often report dissatisfaction with the amount of information provided—some feeling overwhelmed with excessive information, while others believe they receive too little [6]. The annual Kidney Patient Reported Experience Measure (PREM) consistently highlights that CKD patients feel excluded from shared decision-making about their care [7].

Peer education (PE) is a healthcare intervention whereby services enable people with personal experience of illness to provide education and support others with the same illness. Peer education activities may include a number of formalized activities within the provision of services such as teaching, sharing health information, values and behaviours, to help individuals change a health behaviour or inform treatment decision making. PE is based on the premise that people with similar experiences are best placed to support and provide advice about managing their health and illness in others, provide more relatable information from people with similar lived experiences which is empowering, and reinforces learning [8]. PE has been effectively implemented as a health promotion intervention in schools [9], prisons [10,], and chronic illnesses populations, such as, type two diabetes [11] and HIV [12], demonstrating its potential across diverse healthcare settings and cultural contexts. Peer education is seen as a way to involve people in their healthcare who are 'hard

to reach' in existing service delivery infrastructures. In some instances, peer educators and learners are matched on certain background characteristics (e.g., age, sex, ethnicity), and engage with learners in community and home settings rather than healthcare contexts [13]. A peer educator intervention in kidney disease was developed by Kidney Research UK to reach people with chronic kidney disease from diverse ethnic backgrounds [14]. The model was found to be a flexible and adaptable approach to address a range of issues, including early disease detection, organ donation awareness, prevention, management, and end of life issues. More recently, a peer educator model was developed into the ACE (Acceptance, Choice, Empowerment) program in the UK, an award-winning service improvement model of PE designed to support decision making about dialysis options [13,15]. Recent renal guidelines recognise inequity in uptake of home-based dialysis between kidney units and by ethnicity and socio-economic status [16], these patterns are also observed internationally [17]. The ACE peer educator project raised awareness about home-based treatments, was well received by people with CKD, positively evaluated by staff, and seen as impactful by peer educators [13,15]. While developed in the UK context, the principles and components of the ACE program may have relevance for other healthcare systems facing similar challenges in patient education and shared decision-making.

Whilst peer education follows a defined curriculum or intervention model, and typically involves structured teaching, sharing of health information, and influencing behaviour change through knowledge transfer; 'peer support' is a broader, non-directive intervention which involves sharing experiences and understanding, with an emphasis on providing emotional, social, or practical support to help others to cope and adjust to illness, without a formal educational component [18]. Peer mentoring is often used interchangeably with 'peer support' but represents a distinct approach. Unlike peer support, which may be provided in group settings, peer mentoring typically involves a one-to-one relationship between mentor and mentee [17] and may involve working towards a common goal(s), such as adherence to, or improved management of, treatment [19]. To ensure conceptual clarity, our review will focus primarily on peer education interventions (structured, curriculum-based teaching), but will include studies of peer support or peer mentoring where these contain educational components that align with our operational definition. During synthesis, we will clearly distinguish between intervention types and report findings accordingly to minimize conceptual ambiguity (see Table 1). We anticipate these three terms are used synonymously, despite having distinct meanings. To ensure that we don't exclude relevant studies, our search strategy will be developed in a way which is sensitive enough to identify kidney peer support and peer mentoring interventions for assessment against the inclusion criteria.

This scoping review builds on this service innovation work and situates it within the broader international literature on patient engagement and shared decision-making in CKD, to identify and understand the broader literature describing the impact of peer education on patient and clinician outcomes when used as an adjunct to standard kidney pre-treatment education, and to identify the core components that makes a PE intervention successful across diverse healthcare contexts. A search of online databases listing current and existing scoping reviews (PROSPERO, the Cochrane Database and Epistemonikos) found no scoping reviews of peer education in kidney care settings are published, or in process. A recent scoping review by (Elliott et al. [17]) examined peer support interventions (rather than peer education specifically) for people with CKD, highlighting the growing interest in peer-based approaches globally. Our review complements this work by focusing specifically on structured peer education interventions and their role in enabling services to support people's treatment decision-making. The review findings will have broader applicability beyond the UK context, providing insights for healthcare systems internationally that are seeking to understand if and how peer-based education interventions support shared decision making about kidney disease treatment options.

## Review aims and objectives

A scoping review to assess the impact of peer education on treatment decision making for people with chronic kidney disease across diverse healthcare settings will be undertaken by addressing the following objectives to:

**Table 1. Defining the components parts of peer education and peer support.**

| Aspect | Peer Education | Peer Support |
|---|---|---|
| **Primary Role** | Delivery of health education using structured, accurate, and evidence-based knowledge to improve understanding, skills and behaviour change through teaching [9,20] | Provides social support whereby people with lived experience of a condition share knowledge and experiences to support others with similar health problems [21]. |
| **Philosophy** | Lay people are best placed to support and encourage health behaviours in others. Peers are able to assume a level of trust and provide more credible information from a relatable source which allows for open discussion of sensitive topics [22], [8]. [23] | |
| **Mechanisms** | Focuses on the 'what' and 'how' through empowerment, information, learning, and skill-building. | Focuses on 'how it feels' and 'how you cope' providing social support, empathy, connection, and information. |
| **Context** | Healthcare or educational settings. Group education sessions, structured patient pathways, public health campaigns, adherence programmes. | Healthcare or educational settings. Support groups, counselling-style peer programmes, mental health or chronic illness support, one to one mentoring. |
| **Content** | Providing evidence-based knowledge, e.g., treatment options, self-care practices, managing side effects, decision coaching. | Sharing stories, information, coping strategies, decision making, encouragement, validation of feelings. |
| **Structure** | Curriculum- or topic-based; follows learning objectives, often monitored for quality assurance. | Flexible, person-centred; guided by individual needs and emotional state, often monitored for peer/patient experience. |
| **Core Function** | Education, decision making, health promotion, awareness-raising, behaviour change. | Psychological wellbeing, emotional support, decision making, reassurance, reducing stigma, promoting resilience. |
| **Source of expertise** | Delivery of health education; may use teaching aids, decision coaching, guidelines, or structured curricula. | Lived experience of the same condition/situation delivered verbally without additional materials. |
| **Mode of Delivery** | Planned sessions, group teaching, one-to-one structured conversations, workshops, or formal training. Lectures, workshops, demonstrations, group classes, written or digital learning materials. | One-to-one support, peer groups, drop-in sessions, telephone/online chats, informal check-ins. Informal conversations, mentoring, active listening, sharing experiences at the pace of the recipient. |
| **Example in practice** | Teaching a new dialysis patient how to perform fluid balance checks or dietary restrictions. Education/decision coaching about types of dialysis. | Sharing personal experiences of diagnosis, coping, living with kidney disease and/or starting dialysis/receiving transplant. |
| **Training Requirements** | Training in teaching methods, communication skills, and subject knowledge. | Training in safeguarding, boundaries, listening skills. |
| **Peer to peer relationship** | Professionalised and formalised. | More relational and less formal. |
| **Integration with healthcare management** | Reinforces medical information and guidance. | Encourages adjustment to illness and engagement with care. |
| **Challenges to delivery** | - Requires clear limits on scope of teaching.<br>- Risk of delivering incorrect or overly medicalised information without proper oversight.<br>- Requires training and resources for ongoing management and oversight.<br>- Recruitment and retention of peer educators. | - Risk of emotional overburdening.<br>- May blur professional boundaries.<br>- Difficult to measure impact objectively.<br>- Requires training and resources for ongoing management and oversight.<br>- Recruitment and retention of peer educators. |
| **Evidence of effectiveness** | Evidence of effectiveness over a range of health behaviours and health promotion in school, healthcare and prisons, including sexual health, diabetes, HIV/AIDS, smoking, obesity, drug abuse [20]. Improvements mostly seen in knowledge and attitudes with some evidence of health behaviour change [9]. | Quantitative studies show improvements in depression and anxiety, quality of life, and perceived self-efficacy, confidence, knowledge, re-assurance and reduced sense of isolation. Improved measures of dialysis initiation, transplantation processes, and hospitalizations. Qualitative studies report benefits including, reassurance, emotional support, and confidence in treatment decision making [17,24]. |

- Describe models of peer education used to supplement standard pre-treatment education in adults with chronic kidney disease

- Identify specific clinical and patient-reported outcomes impacted by peer education in adults with chronic kidney disease

- Identify the core components of effective peer education interventions for people with chronic kidney disease

- Explore variations in the experience of peer education by patient characteristics and need, with descriptive reporting of findings by ethnicity, socio-economic status, age, gender, and disease stage where data permit.

The review objectives are formulated using the PICO criteria outlined in Table 2.

## Method

### Design

A survey of primary empirical research using a scoping review method. The review is guided by the Joanna Briggs Institute evidence synthesis guidance [25] and reported here using PRISMA-P guidelines (Preferred Reporting Items for Systematic review and Meta-Analysis – Protocols; see S1 [26]. A full version of the protocol is available from lead author (AW).

### Ethical approval

Ethical and research governance approvals are not required for the protocol and scoping review.

### Inclusion and exclusion criteria

The inclusion and exclusion criteria were defined in terms of the population, intervention, comparator, and outcome measures of relevance to the review aims (PICO; [25]).

### Studies will be included if they meet the following criteria

- their primary aim is to examine peer education in a kidney health setting, including primary, secondary and community care.

- include adults (over 18 years of age) with chronic kidney disease (stages 1–5).

- study design includes one of the following: randomized controlled trials (RCTs), including cluster randomized controlled trials; controlled trials, including quasi-randomized trials; controlled before after studies; pilot, feasibility, and cohort studies.

- studies employing qualitative methods.

- compare peer education interventions to standard kidney education.

- published from 2000 onwards

Studies will be excluded if they are non-experimental, case methodologies, discussion, and/or review papers. We will closely review studies that meet the inclusion criteria that include peer support or peer mentoring to assess the components of these interventions and will distinguish between intervention types during synthesis to maintain conceptual clarity.

**Table 2. Research objectives defined using PICO criteria.**

| Component | Definition |
|---|---|
| Population | Studies of adults with chronic kidney disease stages 1–5 |
| Intervention | Peer education provided within kidney care settings (structured, curriculum-based teaching by trained peers) |
| Comparator | Standard pre-treatment education provided within kidney care settings |
| Outcome | Any patient or clinician-reported (health) outcome |

## Search strategy

Search strategies will be developed with reference to the review aims, JBI guidelines [25,27] and an information specialist, to identify articles from online medical and social science databases, including: OvidMedline, PsychInfo (psychology and psychiatry), CINHAL, The Cochrane Central Register of Controlled Trials (CENTRAL) which includes details of published articles taken from bibliographic databases and other published and unpublished sources, Embase which includes pre-prints and conference abstracts, SCOPUS, Web of Science and ClinicalTrials.gov. Grey literature searching will involve: hand-searching key journals; Google Scholar; citation searching; conducting a complete search of reference lists of all articles included in this and prior reviews; key authors will be contacted to request articles. Language restrictions will not be applied to the search. Data will be managed using Covidence software.

Search terms likely to be included are: kidney-related [kidney diseases, organ transplantation, dialysis, haemodialysis, chronic/advanced/end stage and renal/kidney, renal/kidney and transplant/graft] and peer-related [peers, peer counselling, peer tutoring, peer support, peer education, expert patient, peer mentor, peer led programs, peer to peer, patient to patient]. Initial pilot searches found that including terms to define the context of the intervention, i.e., advanced kidney care clinic, pre-dialysis education, transplant education, etc., resulted in few relevant articles being retrieved; broader terms such as 'health education' produced a highly sensitive search retrieving a large volume of studies beyond the scope of this review. Through iterative refinement with an information specialist, we balanced sensitivity (ensuring comprehensive capture of relevant studies) against specificity (minimizing irrelevant retrievals) by using focused kidney-related terms combined with broader peer-related terms, without overly restrictive context modifiers. For an example search strategy see S2 Appendix 1.

## Study selection

All abstracts and titles identified by the search strategies will be evaluated with reference to the review's inclusion and exclusion criteria. The selection process will be piloted by applying the inclusion criteria to a sample of papers to refine and clarify the inclusion criteria and ensure that the criteria can be applied consistently by more than one team member. Studies will be reviewed and managed using Endnote and Covidence software. Studies will be selected in two stages: two reviewers will perform an initial screening of titles and abstracts against the inclusion criteria followed by screening the full text of papers identified as potentially relevant, and where there is no abstract, and when the abstract information was insufficient to make a judgment about inclusion. A decision will be documented for the inclusion of each paper. Lead author (AW) will oversee both stages of screening. Where discrepancy exists, the coders will reach consensus by referring to a third member of the research team (JB).

## Study timeline and dissemination

The search strategies have been developed and data extraction commenced in September 2025, data collection will be completed by December 2025, and we expect to report our findings by March 2026. Data collection forms, data extracted from included studies, and other relevant review materials will be made publicly available using an online data repository. Review findings will be published in a peer-reviewed journal and summarised in a report to inform the development of our planned research project.

## Charting the data

A standardised data extraction form will be consistently applied to all manuscripts that met the inclusion criteria to reduce bias, and piloted on a small sample of papers to improve validity and reliability. The following information will be extracted: general information (title, authors, year location etc.,), study characteristics (aims and objectives, study design, inclusion and exclusion criteria etc.,), participant characteristics (age, sex, ethnicity etc.,), description of intervention and control,

outcome data (unit of analysis, statistical techniques used etc.,) details of intervention group and control, number of participants, outcome measures. See S3 Table 3 for full data extraction form. The form will be piloted on two or three papers initially to ensure it accurately captures relevant data. Data elicited by the form will be managed using Excel spreadsheets. Two reviewers will independently extract all of the information and critically appraise the study quality using the Mixed Methods Appraisal Tool (MMAT) for studies with diverse designs (qualitative, quantitative, and mixed methods) and the Joanna Briggs Institute (JBI) critical appraisal checklists for specific study types (e.g., RCTs, cohort studies). These tools assess methodological quality, risk of bias, and reporting transparency. A flow diagram will be developed to provide details of the study selection process.

## Synthesis

Models of peer education will be summarised and assessed in terms of a) their impact on patient and clinical outcomes reported within the studies organized by outcome categories (clinical outcomes such as treatment uptake, adherence, and hospitalization rates; patient-reported outcomes including knowledge, self-efficacy, quality of life, and satisfaction; and decision-making quality measures) and b) the identification of the core components of peer education, c) an exploration of variations in the experience of peer education by patient characteristics and need. Where data permit, we will descriptively report findings stratified by patient characteristics including ethnicity, socio-economic status, age, gender, and disease stage to identify potential differential effects or implementation considerations. have not pre-defined which patient and clinical outcomes may be included in studies; we anticipate these will include: evaluation of the peer educator and peer learner experience, e.g., knowledge, skills acquisition; decision making quality, health behaviour change and service impact measures. Where study designs and findings are too heterogeneous to integrate statistically, findings will be synthesized using descriptive statistics and text. A summary of findings table will allow comparison of key study information and a descriptive assessment of study quality will be included.

## Discussion

The review findings will describe the way peer education is defined and implemented in kidney services across diverse international healthcare contexts, identifying the goals these interventions are designed to meet, when services involve peers in the education of people with chronic kidney disease and their families. This review will help describe the needs of people to deliver, and engage effectively, with peer education interventions from their differing perspectives (educator, recipient, health professional). This review will capture the different theoretical approaches underpinning peer education interventions, and their likelihood of supporting proactively people with chronic kidney disease and professionals in making shared decisions about future management plans. By situating our findings within the broader literature on patient engagement and shared decision-making in CKD and other chronic conditions, we will identify common principles and context-specific factors that influence implementation and effectiveness. This evidence-base is necessary when identifying the elements needed to implement a complex intervention such as peer education within services to meet multiple decision maker's expectation of effective engagement and will inform policy development for peer education programs internationally. The findings will have implications beyond the UK context. By investigating systematically the active ingredients within peer education interventions across diverse healthcare systems, the review will identify transferable principles and highlight contextual factors (such as healthcare infrastructure, cultural norms, and resource availability) that may influence implementation, and efficacy.

### Study rigour and limitations

To ensure that we obtain high quality, relevant evidence we will adhere to guidance on conducting, reporting, and assessing risk of bias in the studies that we include. We have worked with an information specialist to develop our search strategy using an iterative process, containing multiple keywords, to search seven medical and social sciences databases, and

have modified the strategy to fit the requirements of each database. We will search the grey literature to ensure an extensive search is undertaken and to uncover unpublished data. After de-duplicating our results we will triangulate the data by screening and extracting data using multiple team members. These measures will improve the validity and reliability of the data included in the review. Our scoping review may be limited as we anticipate that there will be a lack of standardised outcome measures reported in trials and analysis may be a challenge, where necessary we will synthesise the findings using simple statistics and qualitative analysis. The review is also limited to studies published from 2000 onwards and those available in databases accessible to the research team, which may exclude some relevant earlier work or publications in less accessible venues.

## Conclusion

This scoping review will synthesize the current evidence from diverse international healthcare contexts and identify gaps in the literature to provide guidance for designing more effective peer education interventions for people with chronic kidney disease globally, with attention to transferability across different healthcare systems and cultural contexts.

## Supporting information

**S1 File.** PRISMA ScR.
(DOCX)

**S2 Appendix.** Search strategy.
(DOCX)

**S3 Table.** Draft extraction form.
(DOCX)

## Author contributions

**Conceptualization:** Anna Winterbottom, Jyoti Baharani, Hilary L Bekker.

**Data curation:** Anna Winterbottom.

**Formal analysis:** Anna Winterbottom, Jyoti Baharani, Hilary L Bekker.

**Investigation:** Anna Winterbottom.

**Writing – original draft:** Anna Winterbottom, Jyoti Baharani, Hilary L Bekker.

**Writing – review & editing:** Anna Winterbottom, Jyoti Baharani, Hilary L Bekker.

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
