## [Decision Letter · Decision Letter 0]

13 Nov 2025

Dear Dr. Winterbottom,

Thank you for submitting your manuscript to PLOS ONE. After careful consideration, we feel that it has merit but does not fully meet PLOS ONE’s publication criteria as it currently stands. Therefore, we invite you to submit a revised version of the manuscript that addresses the points raised during the review process.

We look forward to receiving your revised manuscript.

Kind regards,

Pasyodun Koralage Buddhika Mahesh

Academic Editor

PLOS ONE

Journal Requirements:

When submitting your revision, we need you to address these additional requirements

4. Please include captions for your Supporting Information files at the end of your manuscript, and update any in-text citations to match accordingly. Please see our Supporting Information guidelines for more information: http://journals.plos.org/plosone/s/supporting-information .

Reviewer's Responses to Questions

**Comments to the Author**

1. Does the manuscript provide a valid rationale for the proposed study, with clearly identified and justified research questions?

Reviewer #1: Yes

Reviewer #2: Yes

Reviewer #3: Yes

2. Is the protocol technically sound and planned in a manner that will lead to a meaningful outcome and allow testing the stated hypotheses?

Reviewer #1: Yes

Reviewer #2: Yes

Reviewer #3: Yes

3. Is the methodology feasible and described in sufficient detail to allow the work to be replicable?

Reviewer #1: Yes

Reviewer #2: Yes

Reviewer #3: Yes

4. Have the authors described where all data underlying the findings will be made available when the study is complete?

Reviewer #1: Yes

Reviewer #2: Yes

Reviewer #3: Yes

5. Is the manuscript presented in an intelligible fashion and written in standard English?

Reviewer #1: Yes

Reviewer #2: Yes

Reviewer #3: Yes

You may also provide optional suggestions and comments to authors that they might find helpful in planning their study.

Reviewer #1: 1.I recommend that the authors include the planned data extraction form as a supplementary appendix. While the manuscript outlines the extraction fields in the text, sharing the actual template would greatly enhance methodological transparency and allow future researchers to replicate or adapt the approach more easily.

2.The research questions could be refined to remove some redundancy and achieve sharper focus. Condensing them into three concise questions, addressing (i) the models and contexts of peer education in CKD, (ii) the outcomes these interventions influence, and (iii) the components and implementation factors that drive effectiveness across different patient groups—would streamline the study aims and better align them with the proposed synthesis strategy.

3.A careful editorial review would further improve the clarity and presentation of the manuscript. Minor grammatical issues (for example, replacing “are their variations” with “are there variations”), consistent use of acronyms, shortening overly long sentences, and standardizing reference and DOI formatting would make the text read more smoothly and professionally.

Reviewer #2: Thank you for giving me the opportunity to review this protocol. This is a good study. It's good to use COVIDENCE for this review.

Reviewer #3: Assessing the role of peer education in improving clinical and patient-reported outcomes in adults with chronic kidney disease: a scoping review protocol

This is an interesting protocol that addresses an important issue in chronic kidney disease care. The idea of using peer education to support decision-making and improve patient outcomes is timely and relevant. The manuscript is generally clear and well-structured, but there are several areas where more detail and clarification would strengthen its contribution.

The scope of the review feels somewhat narrow and UK-centric, particularly with the emphasis on the ACE program. It would be helpful to explain how the findings might be generalizable to other healthcare systems or cultural contexts. At present, the global relevance and potential for broader application are not fully clear.

While the manuscript distinguishes between peer education, peer support, and peer mentoring, the inclusion of all three in the search strategy could create conceptual ambiguity. A clearer operational definition for each term and an explanation of how overlapping interventions will be handled in synthesis would improve clarity.

The methodology is well aligned with JBI and PRISMA-P guidance, but some details need elaboration. For example, the search strategy is described in general terms, yet the rationale for including or excluding certain keywords is not fully explained. How will sensitivity and specificity be balanced to avoid retrieving large volumes of irrelevant studies?

The manuscript mentions that quality will be “critically appraised,” but does not specify which tool or criteria will be used. Providing this information would enhance transparency and rigor.

Outcome measures are another area that could be strengthened, while heterogeneity is anticipated, a preliminary framework for classifying outcomes (clinical, patient-reported, decision-making quality) would help guide synthesis.

The discussion section could do more to situate this review within the broader literature on patient engagement and shared decision-making in CKD. Linking the work to global efforts or similar reviews in other chronic conditions would highlight its relevance and originality.

The manuscript mentions variations by ethnicity and socio-economic status but does not explain how these will be analysed or reported. Even if subgroup analysis is not feasible, a plan for descriptive reporting would be valuable.

Overall, this is a promising protocol that addresses an important gap. Strengthening the conceptual framework, clarifying methodological details, and emphasizing generalizability and policy implications will significantly improve its scientific merit and practical utility.

**Do you want your identity to be public for this peer review?** For information about this choice, including consent withdrawal, please see our Privacy Policy

Reviewer #1: No

Reviewer #2: No

Reviewer #3: No

---

## [Author Response · Author response to Decision Letter 1]

12 Dec 2025

Dear Sir/Madam,

RE: Assessing the role of peer education in improving clinical and patient-reported outcomes in adults with chronic kidney disease: a scoping review protocol

Thank you for reviewing our manuscript favourably, we are delighted for the opportunity to address the committee and reviewer’s comments. Each point is addressed in turn below:

Reviewer Comments

Reviewer #1: 1. I recommend that the authors include the planned data extraction form as a supplementary appendix. While the manuscript outlines the extraction fields in the text, sharing the actual template would greatly enhance methodological transparency and allow future researchers to replicate or adapt the approach more easily.

Thank you for this useful comment, we included this table in the appendix .

2.The research questions could be refined to remove some redundancy and achieve sharper focus. Condensing them into three concise questions, addressing (i) the models and contexts of peer education in CKD, (ii) the outcomes these interventions influence, and (iii) the components and implementation factors that drive effectiveness across different patient groups—would streamline the study aims and better align them with the proposed synthesis strategy.

Thank you, we have made minor alterations to the research questions but were concerned that too much refinement would lose some of the specificity of the aims.

3. A careful editorial review would further improve the clarity and presentation of the manuscript. Minor grammatical issues (for example, replacing “are their variations” with “are there variations”), consistent use of acronyms, shortening overly long sentences, and standardizing reference and DOI formatting would make the text read more smoothly and professionally.

Thank you for bringing this to our attention, we have conducted a generalised review of spellings, grammar and sentence length and improved the manuscript throughout.

Reviewer #2: Thank you for giving me the opportunity to review this protocol. This is a good study. It's good to use COVIDENCE for this review.

Thank you for the positive feedback

Reviewer #3: This is an interesting protocol that addresses an important issue in chronic kidney disease care. The idea of using peer education to support decision-making and improve patient outcomes is timely and relevant. The manuscript is generally clear and well-structured, but there are several areas where more detail and clarification would strengthen its contribution.

The scope of the review feels somewhat narrow and UK-centric, particularly with the emphasis on the ACE program. It would be helpful to explain how the findings might be generalizable to other healthcare systems or cultural contexts. At present, the global relevance and potential for broader application are not fully clear.

Thank you for this helpful comment. We have made reference to and described the potential implications of our findings to other healthcare systems and cultural contexts in our introduction and discussion sections, and added in a recent publication (Elliot et al., 2025) ‘highlighting the growing interest in peer-based approaches globally’

While the manuscript distinguishes between peer education, peer support, and peer mentoring, the inclusion of all three in the search strategy could create conceptual ambiguity. A clearer operational definition for each term and an explanation of how overlapping interventions will be handled in synthesis would improve clarity.

To address this point we have added the following sentence: ‘To ensure conceptual clarity, our review will focus primarily on peer education interventions (structured, curriculum-based teaching), but will include studies of peer support or peer mentoring where these contain educational components that align with our operational definition. During synthesis, we will clearly distinguish between intervention types and report findings accordingly to minimize conceptual ambiguity.’

We have also included Table 1, which outlines the different attributes of peer education and peer support. Peer mentoring is included as an attribute of peer support rather than separate intervention as we believe this accurately reflects it’s approach.

The methodology is well aligned with JBI and PRISMA-P guidance, but some details need elaboration. For example, the search strategy is described in general terms, yet the rationale for including or excluding certain keywords is not fully explained. How will sensitivity and specificity be balanced to avoid retrieving large volumes of irrelevant studies?

We have addressed this point with inclusion of the following sentence: ‘Through iterative refinement with an information specialist, we balanced sensitivity (ensuring comprehensive capture of relevant studies) against specificity (minimizing irrelevant retrievals) by using focused kidney-related terms combined with broader peer-related terms, without overly restrictive context modifiers.’

The manuscript mentions that quality will be “critically appraised,” but does not specify which tool or criteria will be used. Providing this information would enhance transparency and rigor.

We have named the appraisal tool that we will include: ‘Two reviewers will independently extract all of the information and critically appraise the study quality using the Mixed Methods Appraisal Tool (MMAT) for studies with diverse designs (qualitative, quantitative, and mixed methods) and the Joanna Briggs Institute (JBI) critical appraisal checklists for specific study types (e.g., RCTs, cohort studies). These tools assess methodological quality, risk of bias, and reporting transparency.’

Outcome measures are another area that could be strengthened, while heterogeneity is anticipated, a preliminary framework for classifying outcomes (clinical, patient-reported, decision-making quality) would help guide synthesis.

Further details about outcome measures and how these will be classified have been added to two sections of the methodology: ‘Outcome measures are another area that could be strengthened, while heterogeneity is anticipated, a preliminary framework for classifying outcomes (clinical, patient-reported, decision-making quality) would help guide synthesis.’

The discussion section could do more to situate this review within the broader literature on patient engagement and shared decision-making in CKD. Linking the work to global efforts or similar reviews in other chronic conditions would highlight its relevance and originality.

The manuscript mentions variations by ethnicity and socio-economic status but does not explain how these will be analysed or reported. Even if subgroup analysis is not feasible, a plan for descriptive reporting would be valuable.

Further detail on how the review sits within the broader literature has ben provided: ‘By situating our findings within the broader literature on patient engagement and shared decision-making in CKD and other chronic conditions, we will identify common principles and context-specific factors that influence implementation and effectiveness. This evidence-base is necessary when identifying the elements needed to implement a complex intervention such as peer education within services to meet multiple decision maker's expectation of effective engagement and will inform policy development for peer education programs internationally.

The findings will have implications beyond the UK context. By examining peer education interventions across diverse healthcare systems, the review will identify transferable principles and highlight contextual factors (such as healthcare infrastructure, cultural norms, and resource availability) that may influence implementation. This will support adaptation of peer education models to different settings globally, while recognizing that implementation strategies may need tailoring to local contexts.’

We have also provided further detail of how ethnicity and socio-economic status will be analysed and reported: ‘Models of peer education will be summarised and assessed in terms of a) their impact on patient and clinical outcomes reported within the studies, organized by outcome categories (clinical outcomes such as treatment uptake, adherence, and hospitalization rates; patient-reported outcomes including knowledge, self-efficacy, quality of life, and satisfaction; and decision-making quality measures) and b) the identification of the core components of peer education, c) an exploration of variations in the experience of peer education by patient characteristics and need. Where data permit, we will descriptively report findings stratified by patient characteristics including ethnicity, socio-economic status, age, gender, and disease stage to identify potential differential effects or implementation considerations.’

Overall, this is a promising protocol that addresses an important gap. Strengthening the conceptual framework, clarifying methodological details, and emphasizing generalizability and policy implications will significantly improve its scientific merit and practical utility.

We are very grateful for this feedback and the opportunity to strengthen the manuscript, thank you.

Yours sincerely,

Anna Winterbottom (on behalf of the co-authors)

---

## [Decision Letter · Decision Letter 1]

18 Jan 2026

Assessing the role of peer education in improving clinical and patient-reported outcomes in adults with chronic kidney disease: a scoping review protocol

PONE-D-25-34226R1

Dear Dr. Winterbottom,

We’re pleased to inform you that your manuscript has been judged scientifically suitable for publication and will be formally accepted for publication once it meets all outstanding technical requirements.

Kind regards,

Pasyodun Koralage Buddhika Mahesh

Academic Editor

PLOS One

Additional Editor Comments (optional):

Reviewers' comments:

Reviewer's Responses to Questions

**Comments to the Author**

1. Does the manuscript provide a valid rationale for the proposed study, with clearly identified and justified research questions?

Reviewer #1: Yes

Reviewer #3: Yes

2. Is the protocol technically sound and planned in a manner that will lead to a meaningful outcome and allow testing the stated hypotheses?

Reviewer #1: Yes

Reviewer #3: Yes

3. Is the methodology feasible and described in sufficient detail to allow the work to be replicable?

Reviewer #1: Yes

Reviewer #3: Yes

4. Have the authors described where all data underlying the findings will be made available when the study is complete?

Reviewer #1: Yes

Reviewer #3: Yes

5. Is the manuscript presented in an intelligible fashion and written in standard English?

Reviewer #1: Yes

Reviewer #3: Yes

You may also provide optional suggestions and comments to authors that they might find helpful in planning their study.

Reviewer #1: 1. The authors have responded carefully and constructively to the previous reviewer feedback, and the revised version is noticeably improved in terms of conceptual clarity, methodological transparency, and scope. The addition of the data extraction form, the clearer definitions of peer education and related concepts, and the expanded discussion of generalisability and equity substantially strengthen the protocol.

2. The distinction between peer education, peer support, and peer mentoring is now clearly articulated and justified, and the inclusion of Table 1 is particularly helpful in preventing conceptual ambiguity. This has strengthened the theoretical coherence of the review and will make it easier for readers to understand what is and is not included in the scope.

3. The review objectives are appropriate for a scoping review and cover relevant dimensions of literature. There is still some conceptual overlap between the objectives relating to “models,” “core components,” and “mechanisms of impact,” and the authors may wish to slightly refine or cross-reference these aims to reduce redundancy, although this is a relatively minor issue.

4. The methodological approach is rigorous and well aligned with current standards for scoping reviews, including the use of JBI guidance and PRISMA-ScR reporting. The search strategy is comprehensive, well justified, and appropriately balances sensitivity and specificity, and the inclusion of grey literature further strengthens the completeness of the review.

5. The use of established quality appraisal tools (MMAT and JBI checklists) is appreciated and enhances the transparency and credibility of the review process, even within the flexible framework of a scoping review.

6. The proposed classification of outcomes into clinical, patient-reported, and decision-making domains is sensible and will support meaningful synthesis across heterogeneous study designs and outcome measures.

7. The planned attention to variation by ethnicity, socio-economic status, age, gender, and disease stage is an important strength of the protocol and reflects current priorities around equity, inclusion, and implementation relevance in healthcare research.

8. The discussion has been improved by situating the review within the broader literature on patient engagement and shared decision-making, and by highlighting the potential relevance of the findings beyond the UK context. This strengthens the manuscript’s contribution and international relevance.

9. A small number of minor editorial and grammatical issues remain (for example, occasional singular/plural inconsistencies and punctuation), and a final careful proofread would further improve clarity and presentation.

10. The note that data extraction has already commenced is potentially worth clarifying in relation to protocol registration and reporting norms, although this is unlikely to undermine the validity of the work.

Reviewer #3: The authors have addressed all the comments provided during the review process and are satisfied with the corrections.

**Do you want your identity to be public for this peer review?** For information about this choice, including consent withdrawal, please see our Privacy Policy

Reviewer #1: No

Reviewer #3: No

---

## [Editor Report · Acceptance letter]

PONE-D-25-34226R1

PLOS One

Dear Dr. Winterbottom,

I'm pleased to inform you that your manuscript has been deemed suitable for publication in PLOS One. Congratulations! Your manuscript is now being handed over to our production team.

Kind regards,

on behalf of

Dr. Pasyodun Koralage Buddhika Mahesh

Academic Editor

PLOS One